# Historical Trends and Driving Forces of River Water Quality Improvement in the Megacity Shenzhen, China

Xiang Sun [1,2,3], Qingping Wu [2], Jiping Jiang [2,*] and Kairong Lin [4,*]

1. School of Environment, Harbin Institute of Technology, Harbin 150090, China; 11949049@mail.sustech.edu.cn
2. School of Environmental Science and Engineering, Southern University of Science and Technology, Shenzhen 518055, China
3. Shenzhen Ruiyang Water Technology Co., Ltd., Shenzhen 518000, China
4. School of Civil Engineering, Sun Yat-sen University, Guangzhou 510275, China
* Correspondence: jiangjp@sustech.edu.cn (J.J.); linkr@mail.sysu.edu.cn (K.L.); Tel.: +86-187-4512-8722 (J.J.)

**Abstract:** The water quality of urban rivers in China has undergone significant improvement since the 13th Five-Year Plan period (2016–2020). Among these, urban rivers in Shenzhen are the most representative. Assessing historical trends and analyzing the driving forces of river water quality improvement is of great importance and provides valuable insights. This study selects two typical watersheds, Maozhou River and Longgang River, to explore how water quality trends link with water control projects in Shenzhen from 2003 to 2020. The historical trends were evaluated using a recently developed index called WQI-DET, which considers DO, COD, $NH_3$-N, TP, and anionic surfactants. Results showed that both rivers were seriously polluted before 2010 and gradually improved during the 12th Five-Year Plan period. After 2010, the water quality improved rapidly thanks to the environmental remediation of the mainstream, especially the interception project of Longgang River around 2010, and the Maozhou River interception project in 2015. The rainwater and sewage diversion renovation project mainly contributed to meeting the standards for Class IV water bodies during the 13th Five-Year Plan period. This study reveals the semi-quantitative link between comprehensive water quality improvement and pollution control engineering measures. It is a helpful review for Shenzhen and provides a useful reference for other cities.

**Keywords:** megacities; WQI; spatial and temporal evolution; trend test; Shenzhen

## 1. Introduction

Rapid urbanization in China has resulted in a growing concentration of population and economy in cities [1]. The United States, Japan, and European countries have spent more than 40 years gradually transitioning from pollution control to sustainable development since 1970s [2,3]. In contrast, urban river water environment management has been a major focus of attention only in recent years for river environmental protection and water engineering in China, therefore necessitating a more rapid and effective improvement of the urban water environment to respond to the requirements of sustainable development [4]. Shenzhen, an early demonstration area for sustainable development, has invested significantly in water treatment projects over the past decade. Through systematic implementation of engineering measures, including constructing sewage collection networks, rainwater and sewage diversion, pollution source correction and clearance in small areas, comprehensive improvement of main and tributary streams and dark culverts, and upgrading and renovating water quality purification plants, river water quality has significantly improved. In fact, the water quality in Shenzhen is now at its best level since monitoring began in 1982. The results have been impressive.

Despite experiencing rapid population and economic growth, Shenzhen's water environment deteriorated drastically in the late 1980s to the 1990s. In the 1980s, the city's rivers could still serve as an auxiliary water source for urban water supply. However, in

the 1990s, several river diversion projects stopped operating due to water quality problems, and the pollution in Shenzhen's rivers became a pressing issue for experts and scholars [5]. As of 2003, the city's population had reached 7 million, surpassing 10 million in 2010, and exceeding 17 million in 2020.

In the pre-2000 period, Shenzhen's river-related planning and design primarily focused on either flood control or environment. However, the treatment of some rivers led to a mismatch between their surroundings and the river's aesthetic. Additionally, construction projects contributed to soil erosion, exacerbated water pollution problems, and reduced the water surface. Out of the city's 310 rivers, 227 of them were polluted. According to Huang Yilong [6] and others, water quality in Shenzhen began to decline after 1990, with major rivers still maintaining Class IV water quality. However, after 1995, industrialization and urbanization led to a rapid degradation in water quality to poor Class V water.

Therefore, it is essential to conduct a scientific analysis and assessment of water treatment in Shenzhen, which can provide valuable insights for future water treatment and quality improvement projects in Shenzhen and other cities. This study focuses on four study sites around the rivers in Shenzhen, the Maozhou River and Longgang River, covering both upstream and downstream. The Maozhou River, which is situated in the northwest region of Shenzhen, was previously recognized as one of the most heavily polluted water bodies in the city, emitting unpleasant odors. The degradation of the surrounding towns in the river basin has resulted from a combination of imprudent resource utilization, rapid population growth in the short term, and a lack of consideration for natural environmental preservation [7]. Similarly, the Longgang River, located in the northwest area of Shenzhen, poses a significant threat to the water quality of the primary Dongjiang river and even extends its impact to Hong Kong, primarily due to the pollution caused by urbanization.

It collects and organizes historical water quality monitoring data, information on treatment projects, and analyzes the historical process of water quality evolution and driving forces. This study provides a quantitative evaluation and summary of Shenzhen's water environment treatment work over the past 20 years, particularly after 2010 during the 12th and 13th Five-Year Plans.

## 2. Research Area and Data

### 2.1. Research Area

Shenzhen is situated in the Pearl River Delta region of Guangdong Province and covers a land area ranging from 113°45′44″ E to 114°37′21″ N and 22°26′59″ N to 22°52′56″ N. Bordered by Daya Bay to the east, the Pearl River Estuary to the west, Dongguan City and Huizhou City to the north, and Hong Kong Special Administrative Region to the south, Shenzhen has a total area of 1948.69 km². The city's terrain generally trends southeast–northwest and comprises hills, plains, low mountains, and terraces. Shenzhen lies south of the Tropic of Cancer, with an average annual temperature of 22.4 °C, close to 2000 mm of annual precipitation, and the occurrence of many typhoons during summer.

Shenzhen boasts a network of 310 rivers, covering a total length of 998.9 km. Within this network, five rivers stand out with watershed areas larger than 100 km², namely Maozhou River, Guanlan River, Longgang River, Pingshan River, and Shenzhen River. For the purposes of this study, our focus will be on Maozhou River and Longgang River, as they represent the most prominent areas of interest. Maozhou River originates from Yangtai Mountain and flows towards the Pearl River Estuary to the northwest, belonging to the Pearl River Delta basin. Meanwhile, Longgang River originates from Wutong Mountain and flows towards the Dongjiang River to the northeast, belonging to the middle and lower reaches of the Dongjiang River basin.

### 2.2. Monitoring Campaign

(1)　Monitoring sections

The Shenzhen Ecological Environment Bureau has been conducting long-term water quality monitoring in the Maozhou River and Longgang River since 2003, using selected

typical cross-sections(Figure 1, Table 1). For the Maozhou River, the upstream section at Loucun S1 section (113.91° E, 22.78° N) was chosen, located at the junction of Guangming District and Bao'an District. This section has a steeper slope drop of 4‰ compared to the downstream section at Yanchuan S2 section (113.85° E, 22.80° N), which has a slower slope drop of 1.5‰. Neither section is affected by the tidal retreat of the Pearl River Estuary. Meanwhile, for the Longgang River, the upstream section at Dishan Village's S3 section (114.29° E, 22.74° N) was selected, with a slope drop of 7‰, while the downstream section at Xiapi S4 (114.35° E, 22.77° N) has a slope drop of 2.8‰.

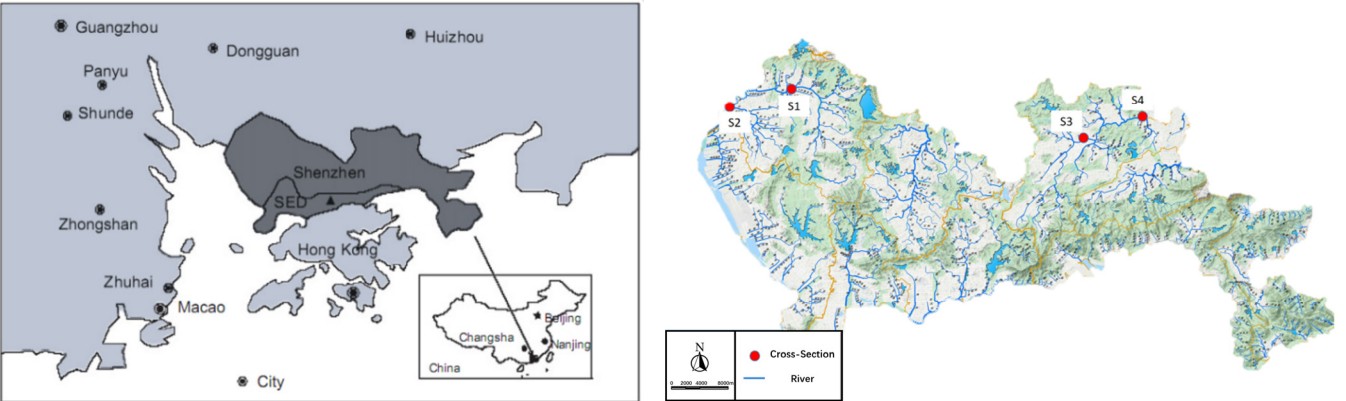

**Figure 1.** Watersheds in Shenzhen and monitoring sites on Longgang River and Maozhou River.

**Table 1.** Basics of the water quality observation sites.

| Rivers | Cross-Section | Number | Longitude (E) | Latitude (N) | Items |
|---|---|---|---|---|---|
| Maozhou River | Loucun | S1 | 113.911352 | 22.78217 | DO, COD, NH$_3$-N, TP, a-SAA |
| | Yanchuan | S2 | 113.849599 | 22.79618 | |
| Longgang River | Dishan Village | S3 | 114.289368 | 22.741019 | |
| | Xiapi | S4 | 114.349678 | 22.767369 | |

(2) Monitoring period and frequency

The study covered 18 years of monthly historical data, from 2003 to 2020, including 540 data of Loucun site (2009~2020), 1055 data of Yanchuan site (2003~2020), 1057 data of Dishan village site (2003~2020), and 1039 data of Xiapi site (2003~2020). The monitoring samples were all hand-sampled in rain-free weather by Shenzhen Ecological Environment Bureau. The data were examined, processed to remove outliers, and then analyzed by SPSS 26.0 for difference tests and by EXCEL for Mann–Kendall trend tests.

(3) Water quality indicators and analysis method

Based on the specific pollution sources within the catchment area (including industrial, domestic, and partially agricultural pollution), this study concentrated on analyzing five key parameters as shown in Table 2: dissolved oxygen (DO, mg/L), chemical oxygen demand (COD, mg/L), ammonia nitrogen (NH$_3$-N, mg/L), total phosphorus (TP, mg/L), and anionic surfactant (a-SAA, mg/L). These parameters were selected due to their ability to provide significant insights into the causes of the black and odorous characteristics observed in urban water bodies, including oxygen availability, organic pollution, nutrient enrichment, and presence of harmful substances.

**Table 2.** Importance of each of the selected variables.

| Variables | Unit | Significance |
|-----------|------|-------------|
| DO | mg/L | It indicates the water's capacity to support aquatic organisms and their ability to carry out essential biological processes. |
| COD | mg/L | It indicates the presence of substances that can consume oxygen and potentially degrade water quality. |
| $NH_3$-N | mg/L | Elevated levels of ammonia can be indicative of organic pollution, excessive fertilization, or improper wastewater treatment. |
| TP | mg/L | Phosphorus is a nutrient that can promote excessive growth of algae and aquatic plants, leading to eutrophication. |
| a-SAA | mg/L | The presence of anionic surfactants in water indicates the input of domestic or industrial wastewater. |

The DO was measured at the sampling point using a YSI multi-parameter water quality monitor, while the COD was measured using the potassium dichromate method (GB 11914-1989). Total phosphorus was measured using the ammonium molybdate spectrophotometry method (GB 11893-1989), ammonia nitrogen was measured using the nano reagent spectrophotometry method (HJ 535-2009), and anionic surfactant was measured using the sugba blue spectrophotometry method (GB 7494-87).

## 3. Methodology

Considering that the water quality was in poor Class V before 2010, the graded evaluation method using the Surface Water Environmental Quality Standard (GB 3838-2002) could not reflect its improvement; this study used M-K trend test [8] and a recently developed Water Quality Index (WQI-DET) [9] and other methods to characterize the water quality changes of Maozhou River and Longgang River from 2003 to 2020.

### 3.1. M-K Trend Test

M-K trend test is a nonparametric test method, similar to the parametric test method detection ability, and its advantages are that the object data do not need to obey a certain probability of distribution, human factors, less interference with outliers, a wide range of detection, and a high degree of quantification. This method is widely used in the field of hydrometeorology [10,11].

The Mann–Kendall trend test was utilized by calculating the trend change statistic of the data series. This involved analyzing the performance of $U_F$ along with its inverse series, $U_B$, and comparing the performance of UF at significant levels ($U_{\alpha(0.05)} = 1.96$, $-U_{\alpha(0.05)} = -1.96$ at significant level $\alpha = 0.05$; At the highly significant level $\alpha = 0.01$, $U_{\alpha(0.01)} = 2.58$, $-U_{\alpha(0.01)} = -2.58$). When $U_F > 0$, the hydrological series showed an increasing trend; when $U_F < 0$, the sequences show a decreasing trend. When $|U_F| < 1.96$, the trend is not significant; $1.96 < |U_F| < 2.56$, the trend is significant; $|U_F| > 2.56$, the trend is highly significant.

### 3.2. Comprehensive Pollution Index of Water Quality (WQI-DET)

The WQI-DET method was proposed by Huang et al. (2019) [12] and used to evaluate the water quality status of rivers. Compared with the original index (0–100) [13], the WQI-DET has a range from $-\infty$ (inferior water quality) to 100 (excellent water quality), where water quality below 0 is worse than V. The scores from 0 to 100 are adapted to the Chinese water quality classification standards for rivers and lakes, known as the "Environmental Quality Standards for Surface Water" (GB3838-2002). These standards are divided into five classes: Class I (excellent), Class II (good), Class III (medium), IV (poor), and V (very poor) five water quality classes. The scores are used to map water quality parameters.

The calculation method is as follows [14]:

$$\text{WQI}_{\text{DET}}^{j} = min(\text{WQI}_{\text{DET\_1}}^{j}, \text{WQI}_{\text{DET\_i}}^{j}, \dots, \text{WQI}_{\text{DET\_n}}^{j}) \qquad (1)$$

$$\mathrm{WQI}^{j}_{\mathrm{DET\_i}} = 100 - \max(0, \frac{C_{ij} - C_i^I}{C_i^V - C_i^I} \times 100) \qquad (2)$$

where $\mathrm{WQI}^{j}_{\mathrm{DET\_i}}$ is the WQI-DET index of a water quality variable i in water sample *j*. $C_{ij}$ is the concentration of a water quality parameter *i* in water sample *j*. $C_i^V$ and $C_i^I$ are the indicators of variable *i* under the criteria of Class I and V water, respectively.

## 4. Results and Discussion

### 4.1. Overall Water Quality of Rivers

Based on the data presented in Figures 2 and 3 and Table 3, it is evident that the upstream S1 Loucun section of Maozhou River fully meets the Class III water standard in 2020 (DO $\geq$ 5 mg/L, COD $\leq$ 20 mg/L, ammonia nitrogen $\leq$ 2 mg/L, total phosphorus $\leq$ 0.2 mg/L, anionic surfactant $\leq$ 0.2 mg/L). Specifically, the levels of DO and anionic surfactant started meeting the standard in 2012, while COD and ammonia nitrogen met the standard in 2018 and total phosphorus in 2020. On the other hand, the downstream S2 Yanchuan section still had relatively poor water quality in 2020, as total phosphorus levels did not meet the standard. DO and COD started meeting the standard in 2018, ammonia nitrogen in 2019, and anionic surfactant in 2017.

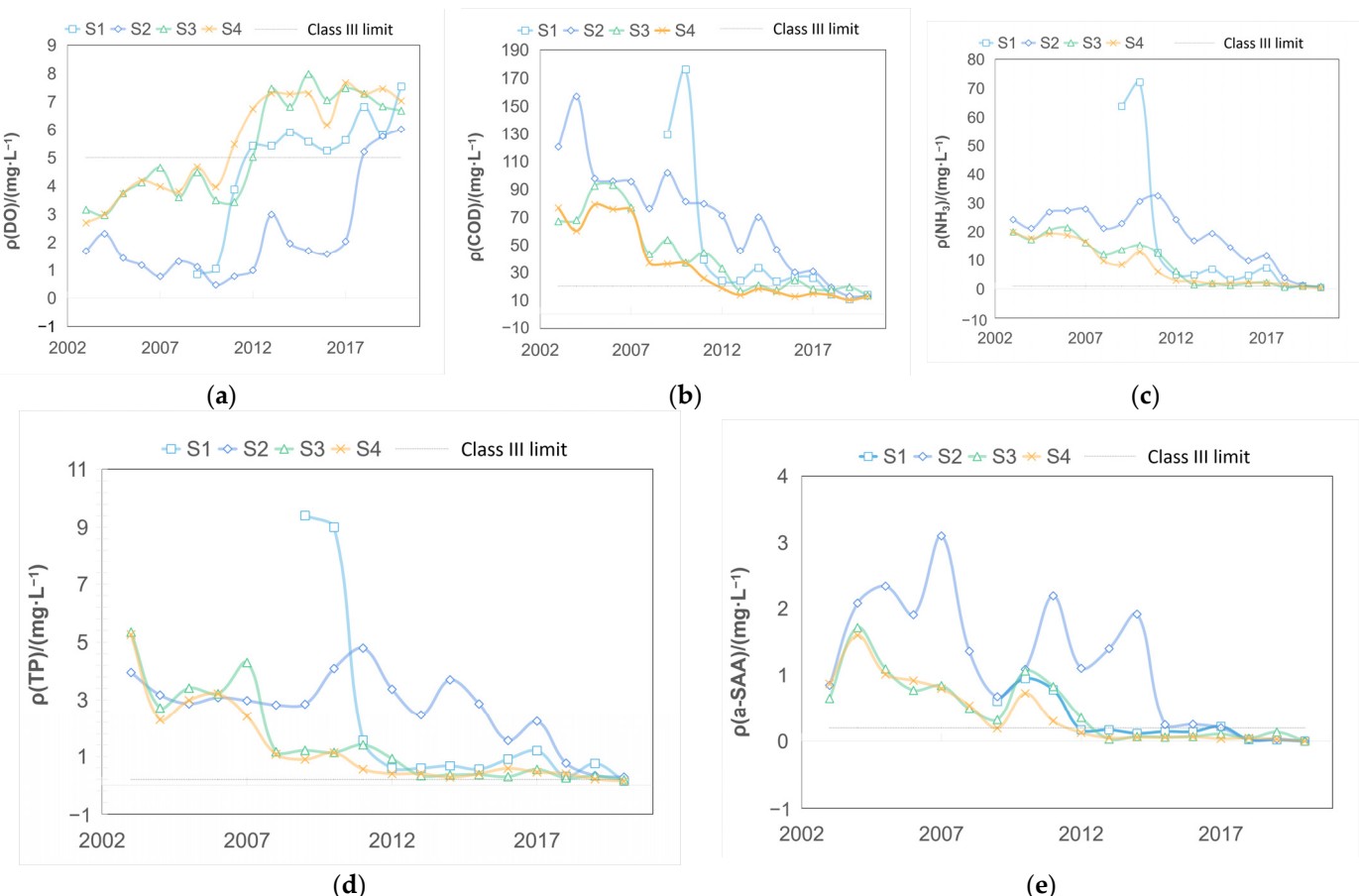

**Figure 2.** Trend of mass concentration of water quality parameters of observation sites. (**a**) DO. (**b**) COD. (**c**) NH$_3$. (**d**) TP. (**e**) a-SAA.

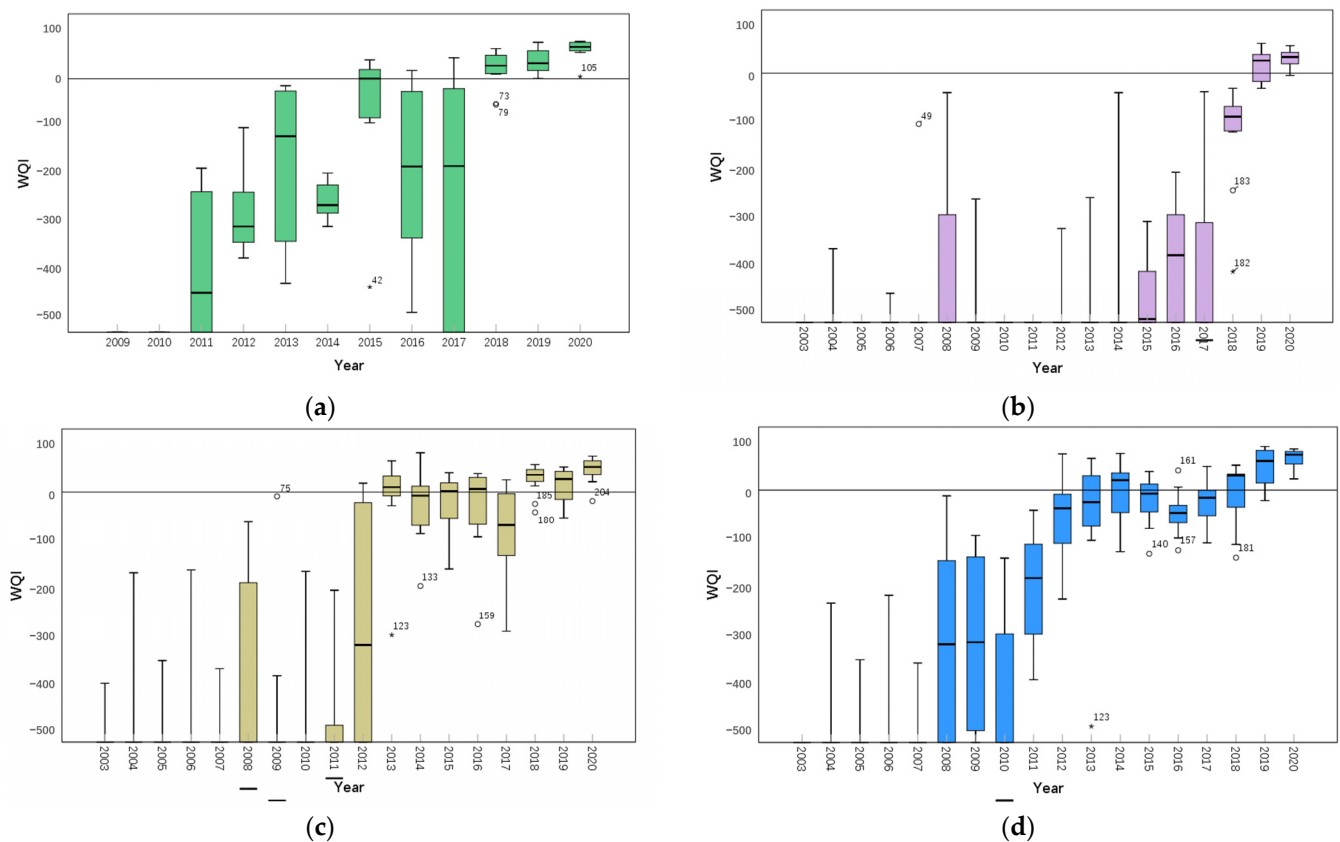

**Figure 3.** Variation of WQI-DET of water quality parameters of observation sites. (**a**) S1 Loucun. (**b**) S2 Yanchuan. (**c**) S3 Dishan Village. (**d**) S4 Xiapi. ○ indicates a discrete value. * denotes extreme values.

**Table 3.** Mass concentration of water quality parameters of observation sites.

| Rivers | Cross-Section | Number | Items | Max (mg/L) | Min (mg/L) | Year of Achieving Class III |
|---|---|---|---|---|---|---|
| Maozhou River | Loucun | S1 | DO | 7.54 | 0.85 | 2012 |
| | | | COD | 176.02 | 10.58 | 2018 |
| | | | NH$_3$-N | 72.1 | 0.46 | 2018 |
| | | | TP | 9.41 | 0.14 | 2020 |
| | | | a-SAA | 0.95 | ND | 2012 |
| | Yanchuan | S2 | DO | 6.02 | 0.77 | 2018 |
| | | | COD | 156.89 | 12.83 | 2018 |
| | | | NH$_3$-N | 30.58 | 0.71 | 2019 |
| | | | TP | 4.8 | 0.28 | 2020 |
| | | | a-SAA | 3.1 | ND | 2017 |
| Longgang River | Dishan Village | S3 | DO | 6.83 | 2.97 | 2012 |
| | | | COD | 93.02 | 13.33 | 2013 |
| | | | NH$_3$-N | 21.43 | 0.88 | 2013 |
| | | | TP | 5.34 | 0.18 | 2020 |
| | | | a-SAA | 1.715 | ND | 2013 |
| | Xiapi | S4 | DO | 7.02 | 2.69 | 2011 |
| | | | COD | 79.03 | 10.13 | 2012 |
| | | | NH$_3$-N | 19.86 | 0.36 | 2017 |
| | | | TP | 5.25 | 0.14 | 2020 |
| | | | a-SAA | 1.598 | ND | 2012 |

For the Longgang River, both the S3 Dishan Village and S4 Xiapi sections fully met the Class III water standard in 2020. In particular, S3 Dishan Village met the standard for DO in 2012, COD and ammonia nitrogen in 2013, total phosphorus in 2020, and anionic surfactant

in 2013. S4 Xiapi met the standard for DO in 2011, COD in 2012, ammonia nitrogen in 2017, total phosphorus in 2020, and anionic surfactant in 2012. Overall, the water quality trend for the typical cross-sections of the Maozhou and Longgang Rivers shows a substantial improvement over the years.

*4.2. Characteristics of Interannual Variation in Water Quality*

To prevent excessive data dispersion and fluctuations in water quality, the WQI-DET trend chart (Figure 3) only shows values of −500 or more on the y-axis. Based on the box plot analysis, the water quality of the typical cross-sections S1–S4 showed an overall improving trend. The WQI-DET value for S1 Loucun cross-section was below −500 before 2010, increased from 2011 to 2015, fluctuated substantially from 2016 to 2017, and remained above 0 after 2018, indicating a consistent improving trend. Similarly, the water quality of S2 Yanchuan cross-section was below −500 before 2015, slightly improved in 2015–2016, fell below −500 again in 2017, significantly improved in 2018, and continued to rise thereafter. The average value of S3 Dishan Village section was below −500 before 2011, improved rapidly in 2012, reached above 0 in 2013, but fluctuated continuously from 2014 to 2016, deteriorated again in 2017, and improved above 0 in 2018–2020, continuing to rise. The water quality of S4 Xiapi section was below −500 before 2007, improved to −300 in 2008–2009, deteriorated again in 2010, and showed an improving trend after 2011, fluctuating above and below 0 in 2015–2017, and reaching above 0 in 2018, continuing to rise thereafter.

Data analysis revealed that the water quality of the upstream section of the Maozhou River was better than that of the downstream section, mainly because the development and construction in the downstream section of the Maozhou River occurred earlier. The Guangming New Area was established above Loucun after 2007, and the Guangming District was established in 2018, with overall development and construction occurring later than in the downstream Bao'an District. Conversely, the water quality of the upstream section of the Longgang River was worse than that of the downstream section, primarily due to the lower concentration and intensity of urban development and construction in the Longgang District than in the Bao'an District. Industries are distributed upstream in Longgang District, contributing to the lower water quality.

*4.3. Increasing Trend of the Composite Water Quality Index*

The trend test of WQI data of the four stations by the Mann–Kendall method showed that (in Table 4) the weekly WQI data of Maozhou River S1, S2, and Longgang River S3, S4 demostrated an increasing trend from 2003 to 2020 over the years, and passed the significance tests of 0.01 and 0.05, respectively, indicating that the increasing trend of WQI is pronounced. This is consistent with the results obtained by regression analysis of year data and water quality data.

**Table 4.** Basics of the water quality observation sites Mann-Kendall test of WQI trend.

| Rivers | Cross-Section | Number | WQI z | WQI Slope |
|---|---|---|---|---|
| Maozhou River | Loucun | S1 | 9.318 *** | 21.904 |
| | Yanchuan | S2 | 9.772 *** | 11.165 |
| Longgang River | Dishan Village | S3 | 5.519 *** | 8.746 |
| | Xiapi | S4 | 38.378 *** | 8.101 |

Note: "***" indicates that the significance test with 99% confidence level was passed.

*4.4. WQI Mutation Test*

The results of the trend test show (Table 4) that the weekly WQI-DET value of Loucun in Maozhou River S1 fluctuated and decreased in 2009, but there has been a significant increasing trend since 2010.This increasing trend passed the 0.05 significance test (($U_{\alpha(0.05)}$ = 1.96) and even passed the 0.01 significance test ($U_{\alpha(0.01)}$ = 2.58) in 2011, which

indicates that the increasing trend of the weekly WQI-DET value of Loucun section is significant. Based on the location of the intersection of the UFk and UBk, it was determined that the increase in the weekly WQI-DET value of Loucun was a sudden change phenomenon. The weekly WQI-DET value of Longgang River S3 Dishan Village section decreased from 2003 to 2004. The trend passed the 0.05 significance test ($p$ value < 0.05) in 2009, 2016, 2018, and 2020. From 2008 to 2011, the decreasing trend passed the 0.01 significance test ($p$ value < 0.01), and there was an increasing trend since 2011. Based on the location of the intersection of the UFk and UBk curves, it was determined that there was a mutation.

The weekly WQI-DET values of the S2 Yanchuan section fluctuated from 2003 to 2004, but the trend was not evident and did not pass the 0.05 significance test ($U_{\alpha(0.05)} = -1.96$), and the trend has been increasing since 2004. The increasing trend passed the 0.05 and 0.01 significance tests in 2015 and rebounded after fluctuating from 2015 to 2016; the overall trend passed the 0.01 significance test. The overall trend passed the 0.01 significance test, which indicates that the increasing trend of the weekly WQI-DET value of the Loucun section is significant and now tends to be stable. Based on the location of the intersection of the UFk and UBk, it was determined that the increase in the weekly WQI-DET value of Yanchuan was a sudden change phenomenon in 2004 and 2020.

The increase in the weekly WQI-DET value of Dishan was a sudden change phenomenon in 2010 and 2011.

Between 2003 and 2004, the Longgang River S4 Xiapi section witnessed a decline in its weekly WQI-DET values, which did not pass the 0.05 significance test. The values fluctuated and increased from late 2004 to 2008 but did not pass the 0.05 significance test. The WQI-DET values decreased between 2008 and 2010, followed by an increase since 2010, with both trends passing the 0.01 significance test. Based on the location of the intersection of the UFk and Ubk, it was determined that the increase in the weekly WQI-DET value of Xiapi was a sudden change phenomenon in 2004, 2009, and 2017.

*4.5. Driving Factor Analysis*

It is generally accepted that the evolution of river water quality is mainly driven by both natural factors and human activities [14]. However, natural factors such as rainfall and temperature have had less influence on the study basin in the last 20 years [15–17]. the average temperature increase rate is 0.338 °C per decade while the precipitation in Shenzhen exhibits a fluctuating pattern characterized by a "rise-fall-rise" trend, with a slight overall decrease in interannual variations over the years, indicating a relatively stable long-term pattern. The river evolution, especially the deterioration of water quality before 2010, is mainly controlled by factors such as population, land use, and industrial development in the urbanization process [6]. Similarly, the rapid improvement of water quality in the last decade was mainly due to the promotion of large-scale water environment management projects in the 12th and 13th Five-Year Plans.

(1) Failure of 'Regional Point Source Intercept' Attempt (2000~2007)

From 2000 to 2007, with the rapid economic and social development, industrial and domestic sewage was discharged into the river, resulting in the gradual deterioration of water quality. The main pollution indexes were poorer Class V and the trend of deterioration. From Figure 4, we can see that the WQI-DET of the Yanchuan section is the lowest, followed by the WQI-DET of Dishan Village and Xiapi, and the WQI-DET of Loucun section is slightly better.

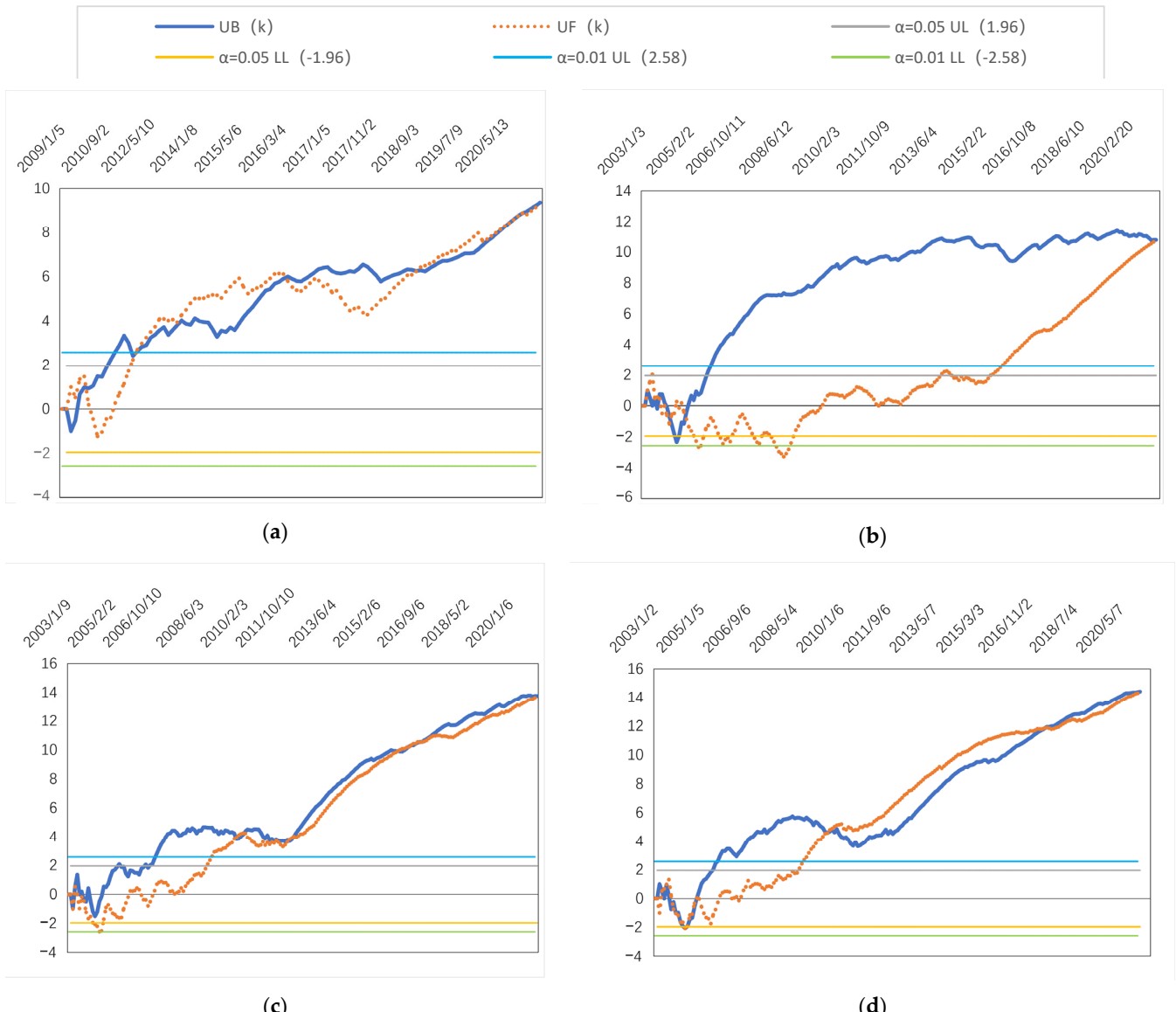

**Figure 4.** Test graph showcases abrupt change of WQI-DET at observation sites. (**a**) S1 Loucun. (**b**) S2 Yanchuan. (**c**) S3 Dishan Village. (**d**) S4 Xiapi.

The main reason for this is that the downstream of Maozhou River is a gathering of industries, and it is a catching tide section, so the pollutants are deposited and the water quality is worse. In contrast, in Guangming District, where the upstream Loucun section is mainly developed by agriculture in the last century, the slope ratio is large, and the flow is fast, the water quality is the best; the industrial gathering in Longgang River basin is lower than the middle and lower reaches of Maozhou River and it is a tributary of Xizhi River, so the overall slope ratio is larger and the capacity of the water environment above the Xiapi section is more significant than that of Dishan Village. Therefore, the difference between the two is not significant.

According to the Shenzhen Bureau of Statistics, the ratio of the three industrial structures in Shenzhen reached 0.1:48.9:51.0 in 2008, with the proportion of tertiary industries exceeded that of secondary industries. This industrial transformation makes a thorough improvement of water quality possible. On the one hand, the number of highly polluting industrial enterprises has decreased, helping to curb the trend of water quality deterioration. On the other hand, in 2002 during the Pearl River work conference in Guangdong Province, the goal for achieving "one year to see the first results, three years not black and

not smelly, eight years the river water becomes clear" was set. In response, the Shenzhen municipal government began to introduce river water quality improvement policy.

In order to swiftly address the issue of sewage pollution in rivers caused by rainwater runoff, the river management in Shenzhen entered the second phase of experimentation while refining the rainwater and sewage diversion system. This phase involved the implementation of a 'point intercept' approach for the sewage within the rainwater pipes. The process entailed identifying the sewage pipe connection that points upstream from the rainwater outlets that discharged sewage. Near these connection points, diversion wells were constructed within the rainwater pipes, redirecting the dry season sewage into nearby sewage pipes.

However, several challenges, such as unstable diversion and rainwater intrusion, emerged from this approach. Practical experience has demonstrated that it is not feasible to achieve comprehensive collection of sewage discharges into the river through the 'point intercept' approach within a short timeframe.

(2)　Diversion sewage Collection and Interception (2008~2015)

In 2011, a 20 km-long sewage box culvert was built along both sides of the river in river remediation project. All the original direct discharge pipes were intercepted into the box culvert or the interceptor along the river and then transported to the sewage treatment plant, which curbed the deterioration of water quality and was the main reason for the sudden change in water quality in 2010 and 2011 in the upstream low mountain village. Therefore, the sudden change point of water quality appeared earlier than upstream in 2009, and the water quality of the mainstream began to improve steadily, which is consistent with the sudden change in test results. However, the capacity of the interceptor box culvert and sewage collection system is limited. With the further development of industries in various districts of Shenzhen, the water quality began to fluctuate due to frequent overflow and leakage.

(3)　Sewage plant upgrade and water reuse (2016~2020)

Shenzhen published "Water Treatment and Quality Improvement Program" in 2016 including targets, strategies, and actions [18]. In 2017, after the large-scale diversion of rainwater and sewage and the strengthening of sewage collection, the sudden change point of water quality appeared again, and the water quality was only able to stabilize the rising channel again. Afterwards, a significantly increased investment was allocated to the upgrading and renovation of wastewater treatment plants (Table 5), leading to further improvement in effluent water quality. This was subsequently utilized as upstream water replenishment for the river.

Time series of regulation progress of Maozhou River and Longgang River is shown in Figure 5. Maozhou River in the upper reaches belongs to Shenzhen City. The left and right banks of the downstream boundary section belong to Shenzhen and Dongguan City. Its remediation, although as early as 2003 by a Shenzhen-Dongguan consultation, has yet to start entirely, because of the project facing acquisition and demolition, coordination, and other issues. The water quality of the upstream changed abruptly in 2009 due to the development and construction brought about by the establishment of the Guangming New Area. The interception and remediation started in 2016, and the effect of rainwater and sewage diversion in 2018–2020 is the main reason for the abrupt change in water quality [19]; the downstream is extremely poor due to the water quality base, so the joint remediation of Shenzhen and Dongguan in 2003 and the rainwater and sewage diversion of the whole basin in 2020 are the main reasons for the abrupt change of water quality. Before 2015, water quality of Maozhou River improved slower than Longgang River. However, in 2015 the vast improvement of the Maozhou River was attributed to lessons learned from the experience of the Longgang River. It was realized that intercepting sewage as the primary engineering effect is limited. Therefore, with an investment of tens of billions of yuan, the full launch of the interception project was implemented, with diversion of rainwater and sewage, as well as source and replenishment projects. Due to the excessive

construction surface, project management, and soil and water conservation problems in 2016 (Figure 6), water quality briefly deteriorated. However, WQI-DET rose rapidly, and water quality was significantly improved after 2018.

**Table 5.** Time series of regulation investment of Maozhou River and Longgang River.

| Watershed | Project | Investment (Million RMB) | | |
|---|---|---|---|---|
| | | The 11th Five-Year Plan (2006–2010) | The 12th Five-Year Plan (2011–2015) | The 13th Five-Year Plan (2016–2020) |
| Maozhou River | River Remediation | 686.5 | 1733.03 | 6163.2 |
| | Sewage Collection | 499.27 | 3192.824 | 362.48 |
| | Sewage Plant Upgrade | 515.75 | 320.05 | 1989.28 |
| | Riverbed Sediment Treatment | 357.41 | -- | -- |
| | water reuse | 0 | 230.44 | -- |
| Longgang River | River Remediation | 0 | 1367.2173 | 1640.54 |
| | Sewage Collection | 150 | 1880.722 | 95.88 |
| | Sewage Plant Upgrade | 588.26 | 88.54 | 1742.7 |
| | Riverbed Sediment Treatment | 285.93 | -- | -- |
| | Water reuse | 0 | 68.43 | -- |

Notes: After 2010, sediment treatment projects were mostly integrated into the river remediation; after 2016, water reuse projects were mostly integrated in sewage plant upgrade.

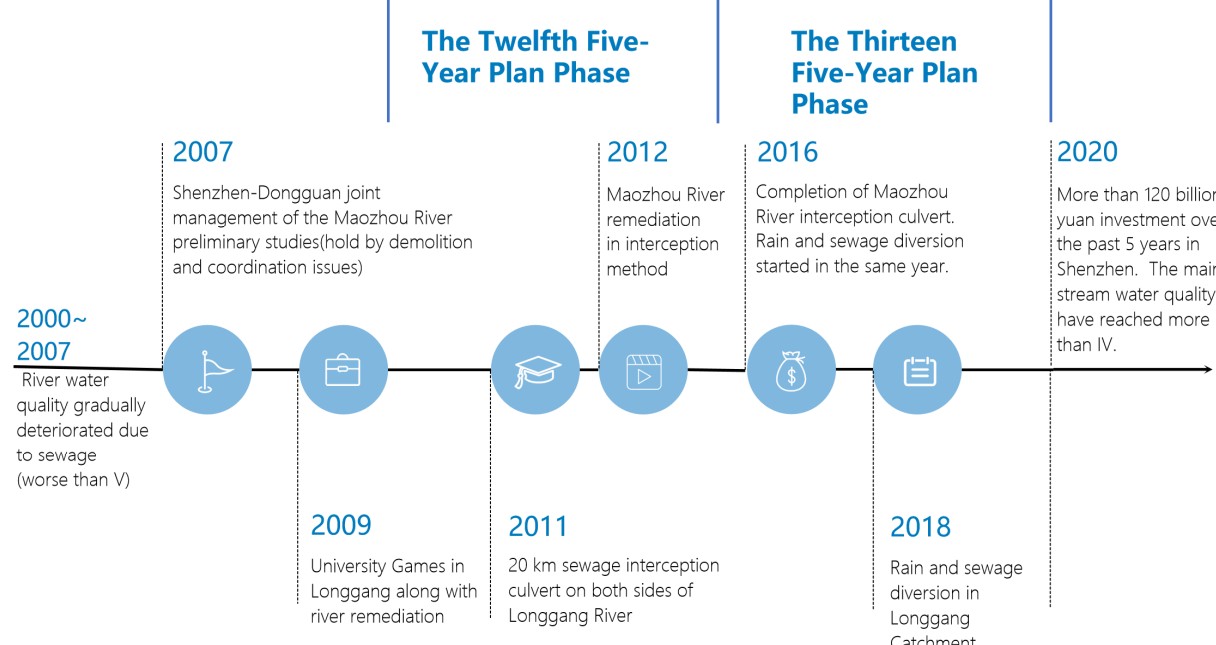

**Figure 5.** Time series of regulation progress of Maozhou River and Longgang River.

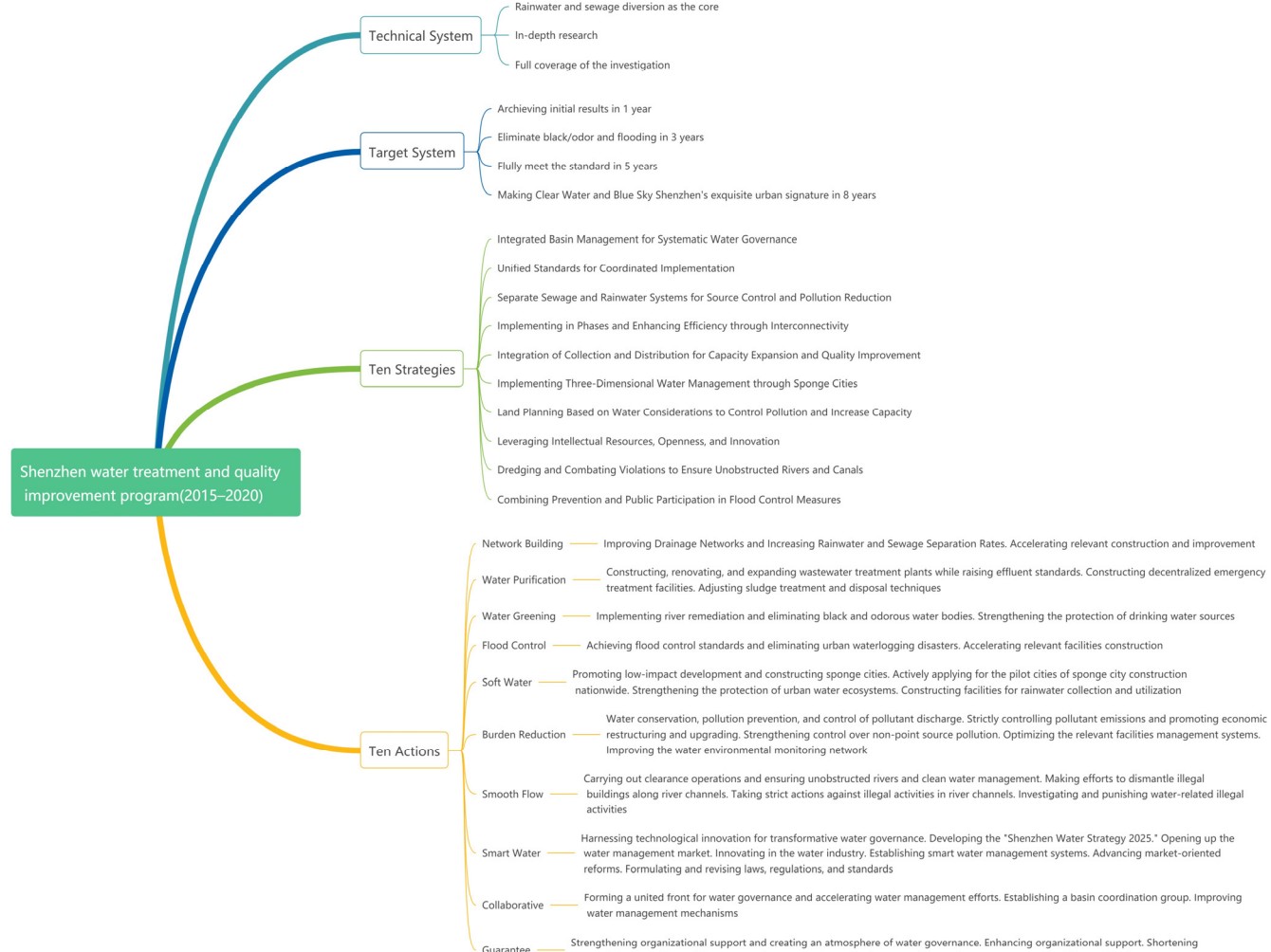

**Figure 6.** Shenzhen Water Treatment and Quality Improvement Program (2015–2020) mind map.

## 5. Conclusions

In this study, based on the water quality monitoring data of Longgang River and Maozhou River in Shenzhen for the past 20 years, a comprehensive assessment of water quality was conducted, and trend analysis and mutation point detection were carried out for the comprehensive water quality index. The driving force analysis of the water treatment project was also conducted, leading to the following conclusions:

(1)　The WQI-DET index method takes into consideration both the current and historical water quality of a body of water, which allows for trend analysis and the identification of patterns in water quality evolution over time. The WQI-DET index method is effective for analyzing the historical process of black odor water bodies, especially in tracking change point when effective actions taken which can be applied to other cities to analyze systematic reviews after black odor treatment.

(2)　The water quality characteristics of the Maozhou River and Longgang River basins differ significantly. Generally, the water quality upstream of Maozhou River is better and more stable. The difference between upstream and downstream water quality in Longgang River is minor, and better than downstream of Maozhou River but worse than upstream of Maozhou River. The improvement trend of Longgang River is more consistent.

(3)　The water quality of Maozhou River has been improving since 2004. However, due to the comprehensive management project that began in 2015–2016, the comprehensive

pollution index has inevitably declined after the rapid improvement and stabilization in 2020.

(4) The trend of water quality changes in Longgang River was not apparent before 2008. Between 2008 and 2011, the comprehensive pollution index first decreased and then increased due to the start of the interception project, and then steadily increased. In 2017, there was a rapid improvement and stabilization due to fluctuations caused by the comprehensive management project.

(5) The main driving factor for the improvement of water quality in Maozhou River and Longgang River is the large-scale water environment improvement project and its sequence. River remediation and interception culverts serve as a defensive line against sewage entering the river in the first place. Subsequently, large-scale sewage collection effectively controls pollution sources. Finally, the water replenishment system of wastewater treatment plants provides the river with a clean water source. After the systematic treatment, the water quality of the two rivers steadily reached the III standard. However, the water quality still cannot meet the standards of surface water class II, due to effluent water quality of the wastewater treatment plant and non-point source pollution.

**Author Contributions:** Conceptualization, X.S. and J.J.; methodology, X.S. and J.J.; data curation, Q.W.; resources, X.S.; writing—original draft preparation, X.S.; writing—review and editing, J.J.; funding acquisition, J.J., investigation, X.S. and K.L. All authors have read and agreed to the published version of the manuscript.

**Funding:** This research was funded by National Natural Science Foundation of China (Grant No. 51979136).

**Data Availability Statement:** Not applicable.

**Acknowledgments:** We thank Shenzhen Water Bureau and Ecological Environment Bureau for their support of the research.

**Conflicts of Interest:** The authors declare no conflict of interest.

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
