# Peer review of "Historical Trends and Driving Forces of River Water Quality Improvement in the Megacity Shenzhen, China"

_water, doi:10.3390/w15122283_

Round 1

Reviewer 1 Report

This study analyzes the results of long-term water quality monitoring in the Shenzhen area of China and discusses the relationship with water quality improvement projects centering on the development of sewage systems. Although the analysis of long-term data is worthwhile, the introduction is very poor and the analysis method is nothing new, and the conclusion that the improvement of sewerage systems contributes to water quality improvement is within the bounds of common sense. In addition, the figures and tables are difficult to read, and there is much room for improvement. For these reasons, I have decided not to accept the manuscript.

Please refer to the specific points raised below for reference when making revisions.

1. Abstract

The paper only introduces the project to improve water quality and does not describe the novelty and usefulness of the research. The last sentence of the conclusion only describes common-sense findings, and it is impossible to determine what has been clarified by this study. The abstract needs to be revised significantly.

2. introduction

Most of the content should have been included in the introduction of the study site, and previous studies were not reviewed. Since there have been many studies on the issue of urbanization and water quality for a long time, please review them appropriately and organize them in a way that readers can understand the necessity and importance of conducting this study.

3. methodology

Since the M-K test is a very commonly used analysis method in time series analysis, the details of the analysis method should be left to the references and a full explanation of the details of the data used should be provided. In particular, it should be described at what spatial and temporal resolution the water quality data were collected, and what processing was used to perform the statistical analysis.

4. results

There are problems with the design and organization of the figures and tables. Major improvements are needed.

5. discussion

Discussion section describes the factors that contributed to the improvement in water quality, but lacks data to support this inference. Background data is needed to increase persuasiveness.

Author Response

General comments:

This study analyzes the results of long-term water quality monitoring in the Shenzhen area of China and discusses the relationship with water quality improvement projects centering on the development of sewage systems. Although the analysis of long-term data is worthwhile, the introduction is very poor and the analysis method is nothing new, and the conclusion that the improvement of sewerage systems contributes to water quality improvement is within the bounds of common sense. In addition, the figures and tables are difficult to read, and there is much room for improvement. For these reasons, I have decided not to accept the manuscript.

AR:Thank you very much for your thorough and constructive review of our manuscript, which we believe will improve the presentation and scientific appeal of the paper. Further, we improve introduction, analysis method, and conclusion. We have also further refined the innovation points,   redraw the figures and tables in the article to ensure that the text can meet the requirements of English publication.

Comments:

Point 1: Abstract-The paper only introduces the project to improve water quality and does not describe the novelty and usefulness of the research. The last sentence of the conclusion only describes common-sense findings, and it is impossible to determine what has been clarified by this study. The abstract needs to be revised significantly.

Response 1: Thank you for pointing this out. We revised abstract to clarify the novelty and usefulness of this study. Please see Lines 12-26,Page 1 in the revised manuscript.

Point 2: Introduction-Most of the content should have been included in the introduction of the study site, and previous studies were not reviewed. Since there have been many studies on the issue of urbanization and water quality for a long time, please review them appropriately and organize them in a way that readers can understand the necessity and importance of conducting this study.

Response 2: Thank you for pointing this out. We have added references and information about the study site into introduction part. Please see reference Lines 445-465, Page 14 in the revised manuscript.

Point 3: methodology-Since the M-K test is a very commonly used analysis method in time series analysis, the details of the analysis method should be left to the references and a full explanation of the details of the data used should be provided. In particular, it should be described at what spatial and temporal resolution the water quality data were collected, and what processing was used to perform the statistical analysis.

Response 3: Thank you for pointing this out. We deleted detailed introduction of M-K test and provide more information on spatial and temporal resolution and processing of the water quality data. Please see reference Lines 151-157, Page 4 in the revised manuscript.

Point 4: Results-There are problems with the design and organization of the figures and tables. Major improvements are needed.

Response 4: Thank you for pointing this out. We redraw the figures and tables in the article to ensure that the text can meet the requirements of English publication.

Point 5: Discussion-Discussion section describes the factors that contributed to the improvement in water quality, but lacks data to support this inference. Background data is needed to increase persuasiveness.

Response 5:  Thank you for pointing this out. We rewrote the discussion chapter by supplementing the project plan and investment data to increase persuasiveness . Please see Lines 265-363,Page 10~11 in the revised manuscript.

Reviewer 2 Report

Dear Authors,

After reading your interesting manuscript, I would like to suggest the following improvements:

1. Include a mind map of the main components of the different phases of the Basin Recovery Plan.

2. Document the set of projects that failed and why.

3. Document the set of projects that were successful and how they could be improved in future plans in other regions.

4. If possible, document the investments made by project and phase of the implemented plan.

5. To make this type of project sustainable, it is important to highlight the water reuse objectives assumed in the different phases of the Recovery Plan and whether these objectives were achieved.

Author Response

General Comments:

After reading your interesting manuscript, I would like to suggest the following improvements:

Point 1: Include a mind map of the main components of the different phases of the Basin Recovery Plan.

Response 1: Thank you for pointing this out. We reviesed figure 5~6 and table 4, detailed basin recovery plan in discussion ,Lines 325-363 in the revised manuscript.

.

Point 2: Document the set of projects that failed and why.

Response 2: Thank you for pointing this out. We added failure 'Point Intercept' projects attempt discussion, Lines 301-312 in the revised manuscript.

Point 3:Document the set of projects that were successful and how they could be improved in future plans in other regions.

Response 3: Thank you for pointing this out. We revised the chapter 3.5, Lines 313-327 in the revised manuscript. 

Point 4: If possible, document the investments made by project and phase of the implemented plan.

Response 4: Thank you for pointing this out. We revised the section 3.5, and added table 4.

Point 5:To make this type of project sustainable, it is important to highlight the water reuse objectives assumed in the different phases of the Recovery Plan and whether these objectives were achieved.

Response 5: Thank you for pointing this out. We rewrote the section 3.5 by supplementing the project plan and investment data including water reuse project to increase persuasiveness . Please see Page 10~11, Lines 330-333 in the revised manuscript.

Reviewer 3 Report

1.     The author did not explain that selecting five variables to measure water quality is sufficient to evaluate the quality.

2.     The author needs to explain the importance of each of the selected variables.

3.     There is no clear indication of the number of samples or measurements that were relied upon for each year of the evaluation period.

4.     More explanations are needed to illustrate why natural factors such as rainfall and temperature have less influenced the study basin in the last 20 years.

5.     On what scientific factor does the author conclude that the WQI-DET index method effectively analyzes the historical process of water bodies?

non

Author Response

Point 1: The author did not explain that selecting five variables to measure water quality is sufficient to evaluate the quality.

Response 1: Thank you for pointing this out. We revised section 2.2 Data Sources, Lines 116-124 in the revised manuscript.

.

Point 2: The author needs to explain the importance of each of the selected variables.

Response 2: Thank you for pointing this out. We reviesed 2.2 Data Sources ,Table 1.

Point 3:There is no clear indication of the number of samples or measurements that were relied upon for each year of the evaluation period.

Response 3: Thank you for pointing this out. We revised 2.2 Data Sources ,Lines 109-115 in the revised manuscript. 

Point 4: More explanations are needed to illustrate why natural factors such as rainfall and temperature have less influenced the study basin in the last 20 years.

Response 4: Thank you for pointing this out. We revised the chapter 3.5, Lines 273-277 by reviewing more references.

Point 5: On what scientific factor does the author conclude that the WQI-DET index method effectively analyzes the historical process of water bodies?

Response 5: Thank you for pointing this out. Using WQI-DET index method, changes in water quality can be evaluated, and necessary actions can be taken to address any issues to improve the water quality.  We revise conclusion. Please see Lines 378-382, Page 10~11,  in the revised manuscript.